# Segregation of complex acoustic scenes based on temporal coherence

**Sundeep Teki[1†]\*, Maria Chait[2†], Sukhbinder Kumar[1,3], Shihab Shamma[4,5], Timothy D Griffiths[1,3]**

[1]Wellcome Trust Centre for Neuroimaging, University College London, London, United Kingdom; [2]UCL Ear Institute, University College London, London, United Kingdom; [3]Institute of Neuroscience, Newcastle University, Newcastle upon Tyne, United Kingdom; [4]The Institute for Systems Research, University of Maryland, College Park, United States; [5]Département d'études cognitive, Ecole Normale Supérieure, Paris, France

**Abstract** In contrast to the complex acoustic environments we encounter everyday, most studies of auditory segregation have used relatively simple signals. Here, we synthesized a new stimulus to examine the detection of coherent patterns ('figures') from overlapping 'background' signals. In a series of experiments, we demonstrate that human listeners are remarkably sensitive to the emergence of such figures and can tolerate a variety of spectral and temporal perturbations. This robust behavior is consistent with the existence of automatic auditory segregation mechanisms that are highly sensitive to correlations across frequency and time. The observed behavior cannot be explained purely on the basis of adaptation-based models used to explain the segregation of deterministic narrowband signals. We show that the present results are consistent with the predictions of a model of auditory perceptual organization based on temporal coherence. Our data thus support a role for temporal coherence as an organizational principle underlying auditory segregation.

**\*For correspondence:** sundeep. teki@gmail.com

[†]These authors contributed equally to this work

**Competing interests:** The authors declare that no competing interests exist.

**Reviewing editor**: Dora Angelaki, Baylor College of Medicine, United States

## Introduction

In our daily lives, we are constantly exposed to complex acoustic environments composed of multiple sound sources, for instance, while shopping in crowded markets or listening to an orchestra. Although we do it effortlessly, the separation of such mixtures of sounds into perceptually distinct sound sources is a highly complex task. In spite of being a topic of intense investigation for several decades, the neural bases of auditory object formation and segregation still remain to be fully explained (*Cherry, 1953*; *McDermott, 2009*; *Griffiths et al., 2012*).

The most commonly used signal for probing auditory perceptual organization is a sequence of two pure tones alternating in time that, under certain conditions, can 'stream' or segregate into two sources (*van Noorden, 1975*; *Bregman, 1990*). Much work using these streaming signals has been carried out to elucidate the neural substrates and computations that underlie auditory segregation (*Moore and Gockel, 2012*; *Snyder et al., 2012*). In a series of seminal experiments, Fishman and colleagues recorded multi-unit activity from the auditory cortex of macaques in response to a simple streaming sequence (*Fishman et al., 2001*; *2004*). For large frequency differences and fast presentation rates, which promote two distinct perceptual streams, they observed spatially segregated responses to the two tones. This pattern of segregated cortical activation, proposed to underlie the streaming percept, has since been widely replicated (e.g., *Bee and Klump, 2004*; *2005*; *Micheyl et al., 2007a*; *Bidet-Caulet and Bertrand, 2009*) and attributed to basic physiological principles of frequency selectivity, forward masking and neural adaptation (*Fishman and Steinschneider, 2010a*). These properties are considered to contribute

**eLife digest** Even when seated in the middle of a crowded restaurant, we are still able to distinguish the speech of the person sitting opposite us from the conversations of fellow diners and a host of other background noise. While we generally perform this task almost effortlessly, it is unclear how the brain solves what is in reality a complex information processing problem.

In the 1970s, researchers began to address this question using stimuli consisting of simple tones. When subjects are played a sequence of alternating high and low frequency tones, they perceive them as two independent streams of sound. Similar experiments in macaque monkeys reveal that each stream activates a different area of auditory cortex, suggesting that the brain may distinguish acoustic stimuli on the basis of their frequency.

However, the simple tones that are used in laboratory experiments bear little resemblance to the complex sounds we encounter in everyday life. These are often made up of multiple frequencies, and overlap—both in frequency and in time—with other sounds in the environment. Moreover, recent experiments have shown that if a subject hears two tones simultaneously, he or she perceives them as belonging to a single stream of sound even if they have different frequencies: models that assume that we distinguish stimuli from noise on the basis of frequency alone struggle to explain this observation.

Now, Teki, Chait, et al. have used more complex sounds, in which frequency components of the target stimuli overlap with those of background signals, to obtain new insights into how the brain solves this problem. Subjects were extremely good at discriminating these complex target stimuli from background noise, and computational modelling confirmed that they did so via integration of both frequency and temporal information. The work of Teki, Chait, et al. thus offers the first explanation for our ability to home in on speech and other pertinent sounds, even amidst a sea of background noise.

---

to stream segregation by promoting the activation of distinct neuronal populations in the primary auditory cortex (A1) that are well separated along the tonotopic axis (*McCabe and Denham, 1997*; *Carlyon, 2004*; *Micheyl et al., 2007a*; *Moore and Gockel, 2012*). Human imaging studies that directly correlated the perceptual representation of streaming sequences with brain responses also support the correspondence between the streaming percept and the underlying neural activity in A1 (*Gutschalk et al., 2005*; *Snyder et al., 2006*; *Wilson et al., 2007*; *Cusack, 2005*). However, similar effects have also been demonstrated in the auditory nerve, suggesting that processes contributing to segregation might occur earlier in the ascending auditory pathway rather than be mediated exclusively by the auditory cortex (*Beauvois and Meddis, 1991*; *Pressnitzer et al., 2008*).

A major drawback of the streaming paradigm is that it uses relatively simple, temporally regular narrowband signals which do not capture the rich spectrotemporal complexity of natural acoustic scenes. Moving beyond streaming, Kidd and colleagues developed a spectrally rich signal referred to as the 'informational masking' (IM) stimulus (*Kidd et al., 1994*, *1995*, *2011*; *Kidd and Mason, 2003*). IM refers to a type of non-energetic or central masking that is associated with an increase in detection thresholds due to stimulus uncertainty and target-masker similarity that is distinct from peripheral energetic masking (*Pollack, 1975*; *Durlach et al., 2003*). These multi-tone masking experiments required listeners to detect tonal target signals in the presence of simultaneous multi-tone maskers, often separated by a 'spectral protection region' (a certain frequency region around the target with little masker energy) that promoted the perceptual segregation of the target from the masker tones. Results demonstrate that target detection is critically dependent on the width of the spectral protection region, and the 'density' of the maskers (*Micheyl et al., 2007b*; *Gutschalk et al., 2008*; *Elhilali et al., 2009b*), and has been hypothesized to rely on the same adaptation-based mechanisms as proposed in the context of simple streaming signals (*Micheyl et al., 2007b*).

In contrast, the sounds we are required to segregate in everyday life are distinct from the narrowband targets used in streaming and IM paradigms; they are often broadband with multiple frequency components that are temporally correlated and overlap with other signals in the environment (*McDermott and Simoncelli, 2011*). Indeed, the ability of models inspired by such paradigms to explain segregation is currently under debate. Recently, Elhilali et al. (2009a) demonstrated that when the two tones in a streaming signal are presented synchronously, listeners perceive the sequence as

one stream irrespective of the frequency separation between the two tones, a result that is inconsistent with predictions based on adaptation-based models. Instead, the authors argued that in addition to separation in feature space, temporal coherence between different elements in the scene is essential for segregation such that temporally incoherent patterns tend to result in a segregated percept while temporal coherence promotes integration (*Shamma et al., 2011*; *Fishman and Steinschneider, 2010b*; *Micheyl et al., 2013a,b*).

To investigate systematically the emergence of an auditory object from a random stochastic background, we developed a new stimulus (Stochastic figure-ground; SFG) consisting of coherent ('figure') and randomly varying ('background') components that overlap in spectrotemporal space and vary only in their statistics of fluctuation (*Figure 1A*; *Teki et al., 2011*). The components comprising the figure vary from trial to trial so that it can be extracted only by integrating across both frequency and time. The appearance of a brief figure embedded in background components thus simulated perception of a coherent object in noisy listening environments. Two spectrotemporal dimensions of the figure were manipulated in each experiment—the 'coherence', or the number of repeating components, and the 'duration', or the number of chords that comprised the figure.

We used psychophysics to examine listeners' ability to extract complex figures and tested segregation behavior in the context of various spectral and temporal perturbations. Our results demonstrate that listeners are remarkably sensitive to the emergence of such figures (*Figure 2*) and can withstand a variety of stimulus manipulations designed to potentially disturb spectrotemporal integration (*Figures 1B–E and 4*). We also show that a model based on the detection of temporal coherence across frequency channels (*Shamma et al., 2011*, *2013*) accounts for the psychophysical data (*Figure 3* and *Figure 3—figure supplement 1*). The work demonstrates an automatic, highly robust segregation mechanism that is sensitive to temporal correlations across frequency channels.

## Results

### Experiment 1: chord duration of 50 ms

In experiment 1, the basic SFG stimulus sequence was used to probe figure-detection performance (*Figure 1A*; see 'Materials and methods'). Listeners' responses were evaluated to obtain d' for each combination of coherence and duration of the figure. The results (*Figure 2A*) show a clear effect of increasing coherence and duration. Hit rates (not shown) mirror d' with listeners achieving mean hit rates of 93 ± 2% for the most salient coherence/duration combination. It is notable that the patterns were very brief (longest figure duration was 7 chords or 350 ms), yet very high levels of performance were observed (and without extensive practice). This is consistent with the idea that this task based on the SFG stimulus taps low-level, finely tuned segregation mechanisms.

### Experiment 2: figure identification

What underlies this sensitivity? Since 'figure-absent' and 'figure-present' signals were controlled for overall number of components (see 'Materials and methods'), a global power increase per se associated with the emergence of the figure, can be discounted as a potential cue. However, it is possible that the decisions of the listeners are based on other changes within the stimulus, for example, the emergence of a figure might be associated with a change in the temporal modulation rate of a few frequency channels. The purpose of experiment 2 was to investigate whether the detection of figures involves a specific figure-ground decomposition, namely whether the figure components are grouped together as a detectable 'perceptual object' distinct from the background components, or whether listeners were rather just detecting some low-level changes within the ongoing stimulus. To address this issue, we created stimulus triplets with different background patterns in which each stimulus contained a figure but where figure components were identical in two out of the three signals. Listeners were required to identify the 'odd' signal that contained a different figure from the other two signals with identical figures in this AXB psychophysical paradigm (see 'Materials and methods' for details). Results (*Figure 2B*) indicate that for the very short figure duration (4 chords, or 200 ms) listeners had difficulty with this discrimination task (d' = 0.31 ± 0.18; not significantly different from 0: p=0.12, t = 1.72), but that performance increased significantly for a figure duration of 8 chords (400 ms; d' = 1.75 ± 0.34) and reached ceiling for a figure duration of 12 chords (600 ms; d' = 2.93 ± 0.26). This pattern of results indicates that figure detection in these stimuli is associated with a segregation mechanism whereby coherent components are grouped together as a distinct perceptual object.

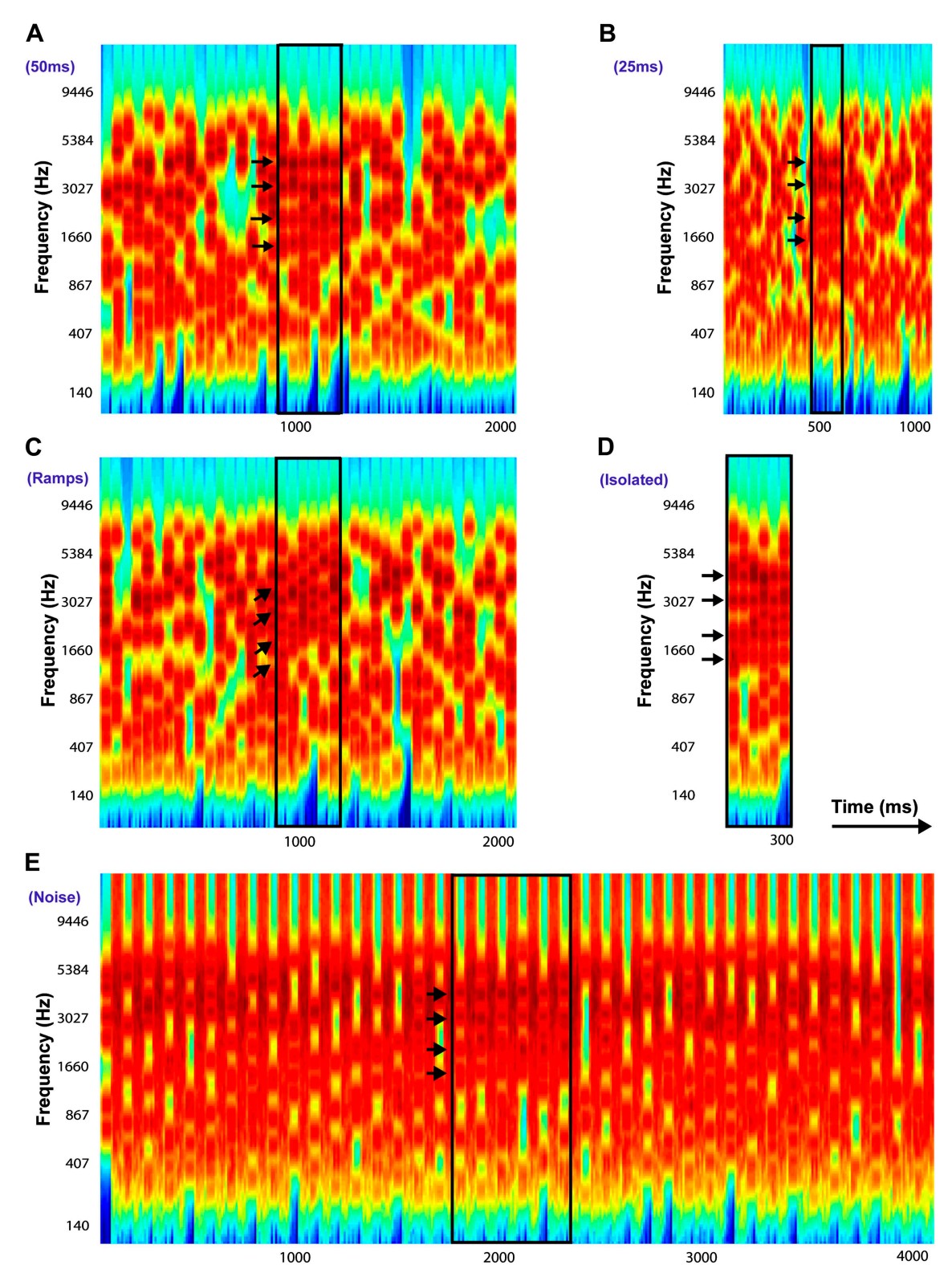

**Figure 1**. Examples of Stochastic Figure-Ground stimuli. All stimuli in this example contain four identical frequency components (only for illustrative purposes: these were selected randomly in the experiments) with $F_{coh}$ = 1016.7 Hz, 2033.4 Hz, 3046.7 Hz, and 4066.8 Hz repeated over 6 chords and indicated by the black arrows. The figure is bound by a black rectangle in each stimulus. (**A**) *Chord duration of 50 ms*: stimulus comprises of 40 consecutive chords each of duration 50 ms with a total duration of 2000 ms. (**B**) *Chord duration of 25 ms*: stimulus comprises of 40 consecutive chords each of duration

*Figure 1. Continued on next page*

*Figure 1. Continued*

25 ms with a total duration of 1000 ms. (**C**) *Ramped figures*: stimulus comprises of 40 consecutive chords each of duration 50 ms each (like **A**) but the frequency components comprising the figure increase in frequency in steps of 2\**I* or 5\**I*, where *I* = 1/24th of an octave, represents the resolution of the frequency pool. (**D**) *Isolated figures*: stimulus comprises only of the 'figure present' portion without any chords preceding or following the figure. The duration of the stimulus is given by the number of chords. (**E**) *Chords interrupted by noise*: stimulus comprises of 40 consecutive chords alternating with 40 chords comprising of loud, masking broadband white noise, each 50 ms in duration. In experiment 6b, the duration of the noise was varied from 100 ms to 500 ms (see 'Materials and methods').

## Temporal coherence modeling

It is difficult to account for listeners' performance in experiments 1 and 2 based on the standard, adaptation-based models proposed in the context of the streaming paradigm (*Micheyl et al., 2007a*; *Fishman and Steinschneider, 2010a*). The figure and background in the SFG stimuli overlap in frequency space, thus challenging segregation based on activation of spatially distinct neuronal populations in A1. Furthermore, the data clearly indicate that performance strongly depends on the number of simultaneously repeating frequency components, suggesting a mechanism that is able to integrate across widely spaced frequency channels, an element missing in previous models based on streaming. Instead, the psychophysical data are consistent with a recently proposed model of auditory object formation based on analysis of temporal coherence (*Shamma et al., 2011*).

The temporal coherence model is based on the idea that a perceptual 'stream' emerges when a group of (frequency) channels are coherently activated against the backdrop of other uncorrelated channels (*Shamma and Micheyl, 2010*). In our stimuli, the 'figure' (defined by the correlated tones) perceptually stands out against a background of random uncorrelated tones. The temporal coherence model postulates that the figure becomes progressively more salient with more correlated tones in the different frequency channels. To measure this coherence, we computed a correlation matrix across all channels of the spectrogram. In principle, the correlation between the activations of any two channels at time *t* should be computed over a certain time window in the past, of a duration that is commensurate with the rates of tone presentations in the channels; this may range roughly between 2 Hz and 40 Hz

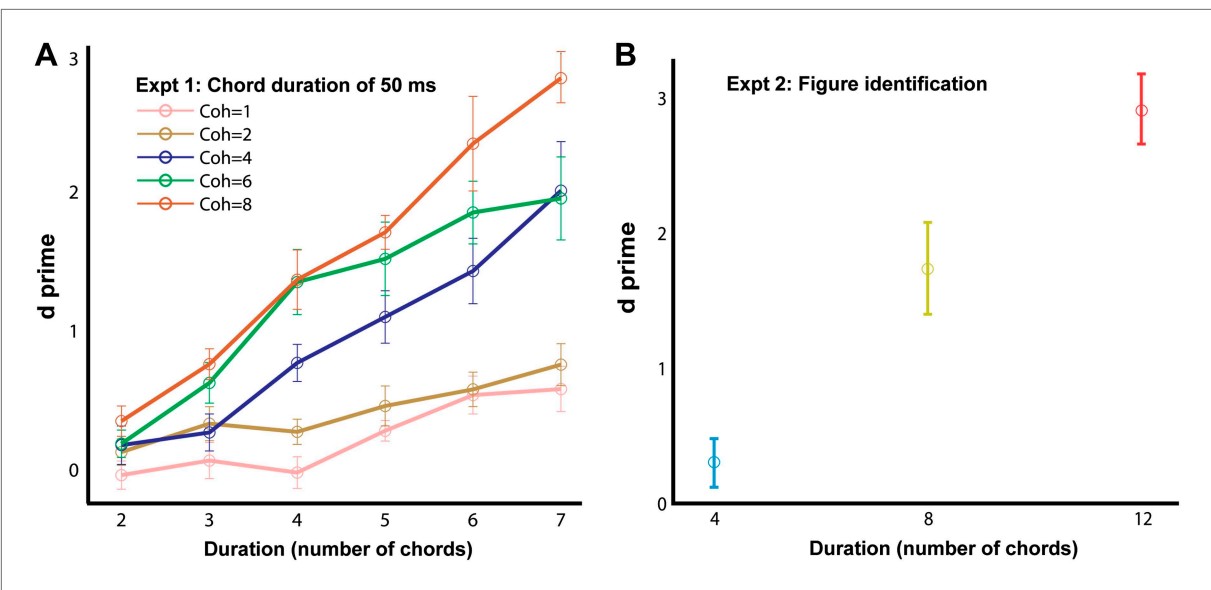

**Figure 2**. Behavioral performance in the basic and figure identification task. The d' for experiments 1 (**A**; 'chord duration of 50 ms'; n = 9) and 2 (**B**; 'figure identification'; n = 9) are plotted on the ordinate and the duration of the figure (in terms of number of 50 ms long chords) is shown along the abscissa. The coherence of the different stimuli in experiment 1 is color coded according to the legend (inset) while the coherence in experiment 2 was fixed and equal to six. The AXB figure identification task was different from the single interval alternative forced choice experiments: listeners were required to discriminate a stimulus with an 'odd' figure from two other stimuli with identical figure components. Error bars signify one standard error of the mean (SEM).

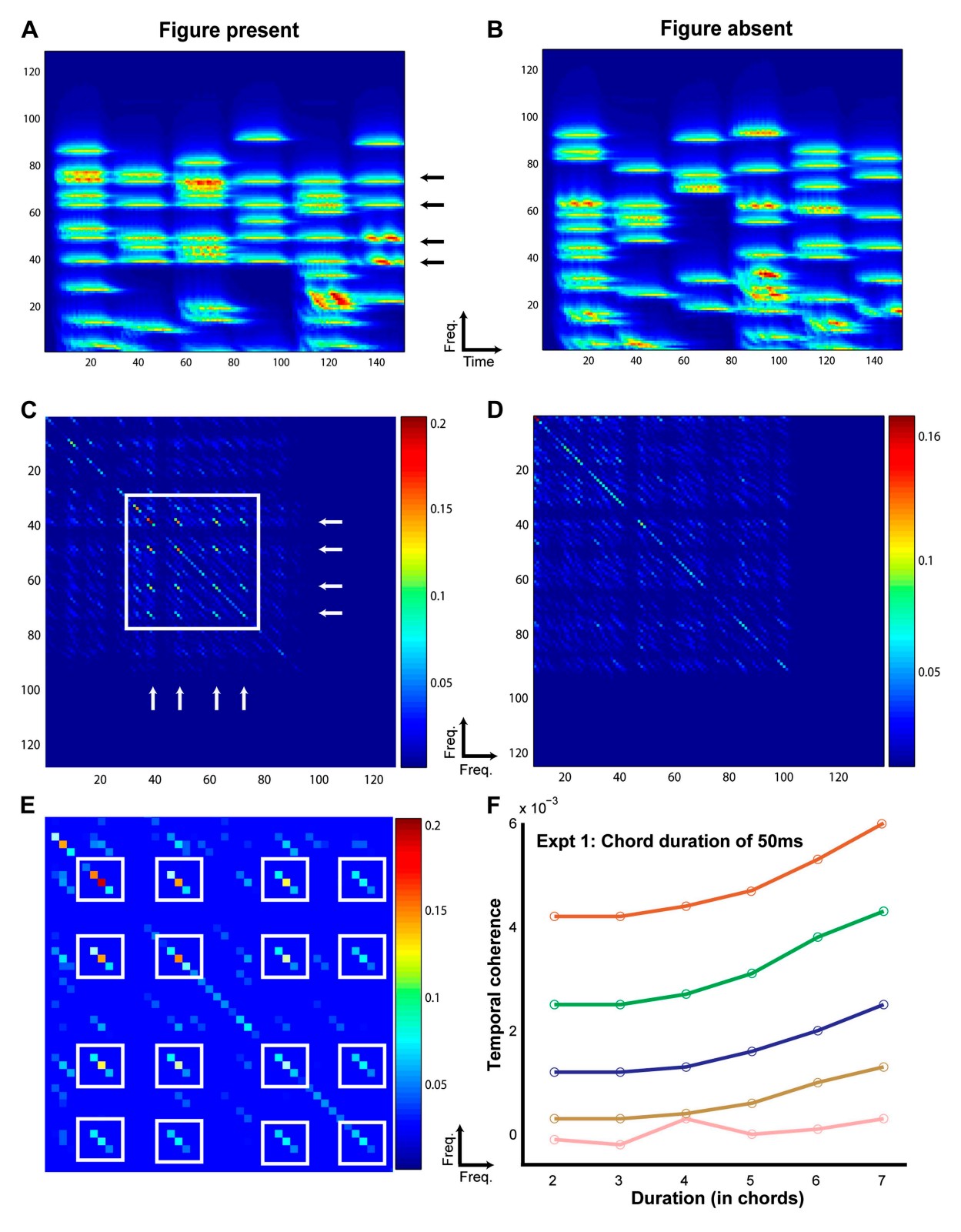

**Figure 3**. Temporal coherence modeling of SFG stimuli. The protocol for temporal coherence analysis is demonstrated here for experiment 5. The procedure was identical for modeling the other experiments. A stimulus containing a figure (here with coherence = 4) as indicated by the arrows (**A**) and another, background only (figure absent) stimulus (**B**) was applied as input to the temporal coherence model. The model performs multidimensional feature analysis at the level of the auditory cortex followed by temporal coherence analysis which generates a coherence matrix for each stimulus as shown in *Figure 3. Continued on next page*

*Figure 3. Continued*

C and D respectively. The coherence matrix for the stimulus with figure present contains significantly higher cross-correlation values (off the diagonal; enclosed in white square) between the channels comprising repeating frequencies as indicated by the two orthogonal sets of white arrows in C. A magnified plot of the coherence matrix for the figure stimulus is shown in E where the cross-correlation peaks are highlighted in white boxes. The strength of the cross-correlation is indicated by the heat map next to each figure. The stimulus without a figure, that is, which does not contain any repeating frequencies, does not contain significant cross-correlations. This process is repeated for 500 iterations ($N_{iter}$) for all combinations of coherence and duration. The differences between these two coherence matrices were quantified by computing the maximum cross-correlation for each set of coherence matrices for the figure and the ground stimuli respectively. Temporal coherence was calculated as the difference between the average maxima for the figure and the ground stimuli respectively. The resultant model response is shown for each combination of coherence and duration in F.

The following figure supplements are available for figure 3:

**Figure supplement 1**. Temporal coherence models for other SFG stimuli.

depending on the experimental session. Consequently, to estimate the perceptual saliency of the figure segment in our stimuli, we computed the correlation matrix simultaneously for a range of temporal resolutions, and then reported the largest correlation values as explained in more detail below and in 'Materials and methods'.

The computations incorporated a spectrotemporal analysis postulated to take place in the auditory cortex (*Chi et al., 2005*). Specifically, temporal modulations in the spectrogram channels were first analyzed with a range of constant-Q modulation filters centered at rates ranging from 2 Hz to 40 Hz (computing in effect a wavelet transform for each channel). The correlation matrix 'at each rate' is then defined as the product of all channel pairs derived from the same rate filters (see 'Materials and methods' and *Figure 3*). The maximum correlation values from each matrix were then averaged and was assumed to reflect the coherence of the activity in the spectrogram channels, and hence the saliency of the figure interval. Note that, as expected, the rate at which the maximum correlations occurred for the different experiments (reported in *Figure 3* and *Figure 3—figure supplement 1*) approximately matched the rate of the tones presented during the figure interval.

*Figure 3* illustrates the modeling procedure and results for stimuli from experiment 1 (see 'Materials and methods' for details of the model). The model successfully accounted for the behavioral data in that, an average cross-correlation based measure was able to systematically distinguish 'figure-present' from 'figure-absent' (or background) stimuli in a manner that mirrored the behavioral responses. The model's measure of temporal coherence showed a similar profile and increased with the coherence and the duration of the figure for the different experimental conditions (*Figure 3—figure supplement 1*). This constitutes the first demonstration of the validity of the temporal coherence model for segregation in complex acoustic scenes.

## Experiment 3: chord duration of 25 ms

In experiment 3, the length of each chord in the SFG stimulus was halved to 25 ms, thereby reducing the corresponding durations of the figure and the stimulus (*Figure 1B*). Here, we aimed to test whether figure-detection performance would be affected by such temporal scaling, that is, whether performance would vary as a function of the total duration of the figure (twice as long in experiment 1 vs experiment 2) or the number of repeating chords that comprised the figure (same in experiments 1 and 2).

Behavioral results (*Figure 4A*) reveal good performance, as in experiment 1. Listeners achieved hit rates of 92 ± 3% for the highest coherence/duration combination used. An ANOVA with coherence and duration as within-subject factors and experimental condition (50 ms vs 25 ms chords) as a between-subject factor revealed no significant effect of condition ($F_{1,15} = 2$; p=0.174), suggesting that performance largely depends on the number of repeating chords irrespective of the time scale. Finally, as expected, model predictions were consistent with the experimental findings. Thus, correlations across the spectrogram channels remained significant, but now occurred at higher rates than in experiment 1 (40 Hz vs 20 Hz), reflecting the faster rate of tone presentations in the figure (*Figure 3—figure supplement 1A*).

## Experiment 4: ramped figures

In the preceding experiments, figure components were identical across several chords. In experiment 4, we manipulated the figure components such that they were not identical across chords but rather

Neuroscience

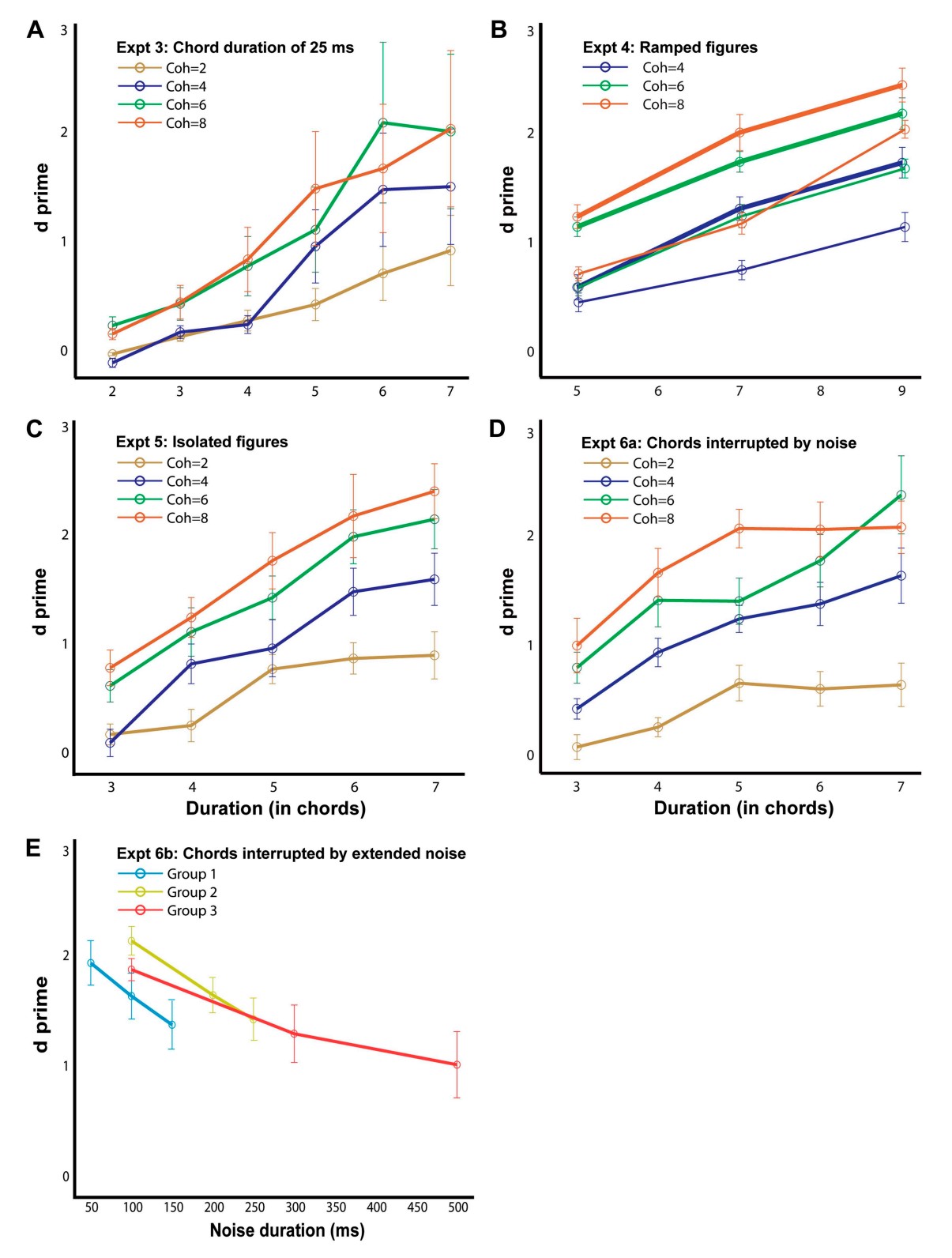

**Figure 4**. Behavioral performance in the psychophysics experiments. The d′ for experiment 3, 4a (thick lines; ramp step = 2), 4b (thin lines, ramp step = 5), 5, 6a and 6b are shown here, as labeled in each figure (n = 10 for all conditions). The abscissa represents the duration of the figure (**A**–**D**) and the duration of the masking noise in **E**. Note that the maximum duration value in experiments 4a and 4b is larger (9 chords) than in the other experiments. Error bars signify one SEM.

ramped, that is, increasing in frequency from one chord to the next (*Figure 1C*). The components in the frequency pool used to generate the SFG signals are separated equally by 1/24[th] of an octave; and in the following two experiments we increased the frequency steps from one chord to the next by two times (experiment 4A; *Figure 4B*—thick lines) or five times (experiment 4B; *Figure 4B*—thin lines) the frequency resolution (i.e., 2/24[th] octave and 5/24[th] octave respectively).

Performance in these experiments was robust (maximum hit-rates of 0.97 and 0.83 were obtained for figures with coherence equal to 8 and duration equal to 7 for the two ramp levels of 2 and 5 respectively) and a comparison with experiment 1 using an ANOVA with coherence and duration as within-subject factors and experimental condition (repeating vs ramp size 2 vs ramp size 5) as a between-subject factor revealed a significant effect of condition: $F_{2,25} = 19$; $p<0.001$. Performance was significantly worse for the ramp = 5 vs ramp = 2 condition ($F_{1,18} = 21$, $p<0.001$), but, remarkably, listeners exhibited above-chance performance even for the steeper slope. This suggests that the segregation mechanisms in question are more susceptible to spectral than temporal perturbations (as in experiments 3, and 6 below) but can still integrate over dynamically changing, rather than fixed, figure components. Finally, as with previous experiments, there were significant correlations among the channels predicting the saliency of the figure. However, the optimal rate at which the correlations occurred here was slightly lower (at 10 Hz; *Figure 3—figure supplement 1B,C*) than that of experiment 1 (20 Hz), perhaps because two 50 ms chords are integrated as a single unit to define the ramp.

## Experiment 5: isolated figures

The stimuli in previous experiments consisted of a sequence of 'background-only' chords, prior to the onset of the figure, and another sequence of 'background-only' chords after figure offset. From first principles, segregation could be considered to be mediated by adaptation to the ongoing background statistics and detection of the figure as a deviation from this established pattern. In order to test this hypothesis, in experiment 5, we removed the 'background' context which preceded the occurrence of the figure (*Figure 1D*; see 'Materials and methods' section). The stimulus consisted simply of the chords which contained a figure (between 3 and 7 chords).

Similar to previous experiments, the results (*Figure 4C*) show a marked effect of coherence and duration, and performance improved with increasing salience of the figures with listeners reaching hit rates of 89 ± 5% for the most salient condition. To evaluate behavior with respect to experiment 1, an ANOVA with coherence and duration as within-subject factors and experimental condition (with background vs no background) as a between-subject factor was used which yielded no significant effect of condition: $F_{1,16} = 0.033$; $p=0.859$, suggesting that the 'background-only' chords which preceded the figure did not affect performance.

Modeling for this experiment replicated the results of experiment 1 in that the correlations increased with the coherence and duration of the figure and showed maximum response at 20 Hz (*Figure 3—figure supplement 1D*), corresponding to the rate of presentation of the chords comprising the figure.

## Experiment 6a: chords interrupted by noise

In experiment 6, we incorporated 50 ms of loud, broadband masking noise between successive 50 ms long SFG chords (*Figure 1E*), in an attempt to disrupt binding of temporally successive components. If figure detection is accomplished by low level mechanisms which are sensitive to a power increase within certain frequency bands, the addition of the noise bursts would disrupt performance by introducing large power fluctuations across the entire spectrum, thus reducing the overall power differences between channels.

The results (*Figure 4D*) show good behavioral performance (maximum hit rate of 0.93 was obtained for the most salient condition) which varied parametrically with the coherence and duration of the figure. An ANOVA with coherence and duration as within-subject factors and experimental condition (50 ms repeating chords vs 50 ms chords alternating with white noise) as a between-subject factor revealed no significant effect of condition ($F_{1,17} = 0.004$; $p=0.953$). Interleaving the noise bursts between successive chords does not therefore affect performance.

Model predictions in this experiment (*Figure 3—figure supplement 1E,F*) are broadly consistent with the findings in that detection became easier with more coherent tones, and with longer figure intervals. The reason is simply because the noise weakens but does not eliminate the correlation among the tones, at least when computed at slower rates.

## Experiment 6b: chords interrupted by extended noise

A natural question that arises from the preceding experiment is—what are the temporal limits or the range over which such a higher-order mechanism operates?

In order to test this question, we gradually varied the duration of the intervening noise bursts between stimulus chords in a set of three related experiments with different durations of noise for a particular combination of coherence (6) and duration (6; see 'Materials and methods'). Results (*Figure 4E*) indicate robust performance for all durations of noise up to 300 ms and surprisingly, supra-threshold performance (d' = 1.00 ± 0.30; significantly different from 0: p=0.01; t = 3.29) for a noise duration of 500 ms. This remarkable ability of listeners to integrate coherent patterns over 3 s long (in the case of 500 ms noise bursts) suggests that the underlying higher-order mechanisms are very robust over such long time windows. Model predictions of these findings are still possible if correlations are measured over longer windows (or slower rates—e.g., 3.33 Hz as in *Figure 3— figure supplement 1F*).

Temporal windows of integration, as long as 500 ms, have rarely been reported in the context of auditory object formation in complex scenes such as those examined here. The results suggest the existence of a central mechanism that is not affected by interfering broadband noise that integrates repeating pure tone components as belonging to a distinct object over multiple time scales. The long temporal windows here implicate cortical mechanisms at or beyond primary auditory cortex (see e.g., *Overath et al., 2008* demonstrating a range of 'cortical windows' between 20 ms and 300 ms).

## Discussion

We demonstrate fast detection with minimal training of a novel figure-from-ground stimulus comprising an overlapping figure and ground segment where, like natural stimuli, the figure has multiple components that are temporally coherent: they start and stop together. We show that such figures can be detected as objects that can be distinguished, consistent with a high-level mechanism for object detection as opposed to the simple detection of frequency or modulation cues (experiment 2). The mechanism scales in time, in that detection depends on the number of components over time rather than their absolute duration (experiment 3) and can still operate if the object is defined by slowly changing frequency trajectories rather than a fixed set of frequencies (experiment 4). It is robust to interruption of the figure by interleaved noise (experiment 5) and can operate over a remarkably long timescale (experiment 6). Modeling based on temporal coherence demonstrated a similar profile to the behavioral results and constitutes the first demonstration of the compatibility of the temporal coherence model with segregation in complex acoustic scenes.

### Segregation based on temporal coherence

The temporal coherence model proposes that segregation is determined not only based on separation in feature-space but rather by the temporal relationship between different elements in the scene, such that temporally coherent elements are grouped together, while temporally incoherent channels with independent fluctuation profiles are perceived as belonging to separate sources (*Shamma et al., 2011*). Specifically, the model incorporates two stages: firstly, a feature analysis stage that performs multidimensional feature analysis by distinct populations of neurons in the auditory cortex that are tuned to a range of temporal modulation rates and spectral resolution scales. Auditory features such as pitch, timbre and loudness are computed by different neuronal groups at this initial stage, the output of which is fed to a second stage that involves analysis of temporal coherence.

Elhilali et al. carried out work implicating a critical role for temporal coherence in the assignment of common elements within a stream: they showed that a pair of synchronous repeating tones produces the same coherent pattern of modeled central activity as a single stream irrespective of the frequency separation between them, suggesting that temporal coherence is an important factor governing segregation (*Elhilali et al., 2009a*). This was substantiated by direct neurophysiological recordings from ferret auditory cortex which showed that synchronous and alternating cortical responses were equally segregated despite their perceptual differences, and hence that the temporal factors are more important in inducing the one and two streams percept (*Shamma and Micheyl, 2010*; *Pressnitzer et al., 2011*; *Shamma et al., 2011*, *2013*).

In the case of the more complex SFG stimuli, the modeling results suggest that temporal coherence is modulated as a function of the coherence and the duration of the figure in a manner similar to the modulation of figure detection performance. Although this is not causal evidence in favor of the model,

it behaves similar to human listeners in complex acoustic conditions as used here. The data suggest temporal coherence as a correlate of stimulus salience by which the brain picks out the most important sounds in busy auditory scenes: a process that may not be computed by dedicated structures but could be achieved by binding across distributed feature channels without significant changes in ensemble activity. Similar accounts of binding in vision based on coherence of the temporal structure have been put forward previously (e.g., *Sporns et al., 1991*; *Alais et al., 1998*; *Blake and Lee, 2005*).

## Neural bases of temporal coherence analysis

It is still not known how temporal coherence may be computed and which brain areas perform these computations. Temporal coherence may be implemented by neurons that show strong sensitivity to temporal coherence across distant frequency channels, or by neurons that act as multiplexers and are more selective to particular combinations of inputs (*Elhilali et al., 2009a*; *Shamma et al., 2011*). Elhilali et al. (2009a) sought such cells in the primary auditory cortex of the ferret but were unable to demonstrate any in passively listening animals but have preliminary evidence in behaving ferrets (*Shamma et al., 2013*).

Although previous brain imaging studies have identified activity in A1 that was correlated with the streaming percept (e.g., *Gutschalk et al., 2005*; *Snyder et al., 2006*; *Wilson et al., 2007*), we found no evidence of modulation of BOLD signal in A1 as a function of figure emergence in a passive listening paradigm (*Teki et al., 2011*). However, we found activity in the intraparietal sulcus (IPS) to be strongly modulated by the salience of the figure, similar to the modulation of temporal coherence observed here. The IPS activation likely reflect bottom-up stimulus-driven processing of figures and is consistent with accumulating literature which suggests that areas outside the conventional auditory system, such as the parietal cortex may have a role in auditory segregation (*Cusack, 2005*; *Dykstra et al., 2011*). Although not relevant to the passive fMRI experiment (*Teki et al., 2011*), attention also influences segregation. In this regard, the parietal cortex is in an ideal position to integrate both bottom-up auditory input as it receives auditory input from the temporoparietal cortex (*Pandya and Kuypers, 1969*; *Divac et al., 1977*; *Hyvärinen, 1982*; *Cohen, 2009*) as well as top-down attentional input from the prefrontal cortex (*Andersen et al., 1985*; *Barbas and Mesulam, 1981*; *Petrides and Pandya, 1984*; *Stanton et al., 1995*). IPS is associated with both bottom-up and top-down attention and is a key structure implicated in saliency map models of visual search (*Koch and Ullman, 1985*; *Itti and Koch, 2001*; *Walther and Koch, 2007*) where low-level feature maps may combine with top-down cognitive biases to represent a global saliency map (*Gottlieb et al., 1998*; *Geng and Mangun, 2009*; *Bisley and Goldberg, 2010*). IPS (and its monkey homologue) has been implicated in mediating object representations, binding of sensory features within and across different modalities, as well as attentional selection.

We hypothesize that IPS may represent a neural correlate of the figure percept where the representation will depend on the salience of auditory figures. In our model, this perceptual representation depends on the computation of temporal coherence across multiple frequencies that are initially represented in the auditory cortex. Neurophysiological recordings from parietal neurons might in future determine whether such sensory analysis (before perceptual representation) involves parietal neurons or is established in auditory cortex first.

## Materials and methods

### Stochastic figure-ground stimulus

We developed a new stimulus (Stochastic figure-ground [SFG] stimulus; *Teki et al., 2011*) to model naturally complex situations characterized by a figure and background that overlap in feature space that are only distinguishable by their fluctuation statistics. Contrary to previously used signals, the spectrotemporal properties of the figure vary from trial to trial and the figure can only be extracted by binding the spectral components that comprise the figure across frequency and time.

*Figure 1* presents examples of the SFG stimulus which consists of a sequence of random chords, each 50 ms in duration with 0 ms inter-chord-interval, presented for a total duration of 2000 ms (40 consecutive chords). Each chord contains a random number (average: 10 and varying between 5 and 15) of pure tone components. The spectral components are randomly selected from a set of 129 frequencies equally spaced on a logarithmic scale between 179 Hz and 7246 Hz such that the separation between successive components is 1/24[th] of an octave. The onset and offset of each chord

are shaped by a 10 ms raised-cosine ramp. In half of these stimuli, a random number of tones are repeated across a certain number of consecutive chords (e.g., in *Figure 1*, four components marked by arrows are repeated across 6 chords) which results in the percept of a 'figure' that readily pops out of the random tonal background. To eliminate correlation between the number of figure and background components, the figure was realized by first generating the random background and then adding additional, repeating components to the relevant chords. To avoid the problem that the interval containing the figure might, on average, also contain more frequency components, and to prevent listeners from relying on this feature in performing the figure detection task, the remaining 50% of the stimuli (those containing no figure) also included additional tonal components, which were added over a variable number (2–7) of consecutive chords at the same time as when a figure would have appeared. But these additional components changed from chord to chord and did not form a coherent figure.

In the present study, we parametrically varied the number of consecutive chords over which the tones were repeated ('duration') and the number of repeated frequency components ('coherence'). The onset of the figure was jittered between 15 and 20 chords (750–1000 ms) post stimulus onset.

## Participants

All participants tested in this set of experiments reported normal hearing and had no history of audiological or neurological disorders. Experimental procedures were approved by the research ethics committee of University College London (Project ID number: 1490/002), and written informed consent was obtained from each participant. For each experiment we report the number of listeners whose data is included in the final analysis. In each experiment, a few listeners (2–3) were excluded from analysis because of their inability to perform the task. 9 listeners (2 females; aged between 20 and 47 years; mean age: 26.9 years) took part in experiment 1. 9 listeners (6 females; aged between 22 and 28 years; mean age: 23.8 years) participated in experiment 2 based on the AXB design. 10 listeners (5 females; aged between 20 and 36 years; mean age: 25.7 years) took part in experiment 3. 10 listeners (5 females; aged between 23 and 31 years; mean age: 26.8 years) participated in experiment 6a. 27 listeners (Group 1: 9 listeners; 5 females, aged between 19 and 27 years; mean age: 21.1 years; Group 2: 10 listeners; 3 females; aged between 19 and 25 years; mean age: 21.3 years; Group 3: 8 listeners; 3 females; aged between 19 and 29 years; mean age: 22.4 years) participated in experiment 6b. 10 listeners (6 females; aged between 21 and 34 years, and mean age of 24.7 years) participated in experiment 4a with ramp step equal to 2 and another group of 10 listeners (3 females; aged between 20 and 30 years and mean age of 24.5 years) took part in experiment 4b with ramp step of 5. 10 listeners (5 females; aged between 22 and 31 years, mean age: 24.8 years) participated in experiment 5.

## Stimuli

SFG stimuli in experiment 1 consisted of a sequence of 50 ms chords with 0 ms inter-chord interval and 2 s duration (40 consecutive chords). The coherence of the figure varied between 1, 2, 4, 6 or 8 and the duration of the figure ranged from 2 to 7 chords. Stimuli for all combinations of coherence and duration were presented in a separate block (total of 30 blocks) where 50% of the trials (50 trials per block) contained a figure.

The stimuli in experiment 2 consisted of 50 ms chords and a figure coherence value of 6. Figure duration varied between 4, 8 and 12 (in separate blocks). Stimuli, all containing a figure, were presented in triplets as in an AXB design (e.g., *Goldinger, 1998*). The background patterns were different in all three signals but two of them (either A and X or B and X) contained identical figure components. Listeners were required to indicate the 'odd' figure (A or B) by pressing a button. Three blocks of 60 trials each were presented for each duration condition.

Stimuli in experiment 3 were identical to those in experiment 1 except that chord duration was reduced to 25 ms. The coherence of the figure varied between 2, 4, 6 or 8 and the duration of the figure ranged from 2–7 chords resulting in a total of 24 blocks.

In experiment 4a and 4b, stimuli were similar to those in experiment 1 except that in this condition, the successive frequencies comprising the figure were not identical from one chord to the next but increased across chords in steps of $2*I$ or $5*I$, where $I = 1/24$th of an octave is the resolution of the frequency pool used to create the SFG stimulus. The coherence of the figure was 4, 6, or 8 and duration was 5, 7 or 9 chords resulting in a total of 9 blocks for each condition. Note that in this experiment, the maximum duration of the figure (9 chords) is longer than the maximum duration of the figure in the remaining experiments (7 chords).

The stimuli for experiment 5 were same as in experiment 1 except that they comprised of the figure only (3–7 chords or 150–350 ms) without any chords that preceded or succeeded the figure as in previous experiments. The coherence of the figure was 2, 4, 6, or 8 chords and this resulted in a total of 20 blocks.

For experiment 6a, we modified the SFG stimulus so that successive chords were separated by 50 ms broadband noise burst. The loudness of the noise was set to a level 12 dB above the level of the stimulus chords. The coherence of the figure was 2, 4, 6 or 8 and the duration of the figure ranged from 3 to 7 chords resulting in a total of 20 blocks.

The stimuli in experiment 6b were identical to the previous experiment save for the following differences: (a) coherence and duration were fixed at a value of 6; (b) the duration of the noise was varied in three different experiments in increasing order: group 1: 50, 100, 150 ms; group 2: 100, 200, 250 ms; group 3: 100, 300, 500 ms respectively. The 100 ms condition was chosen as an anchor and only those participants who performed above a threshold of d' = 1.5 in this condition were selected for the whole experiment.

## Procedure

Prior to the study, training was provided which consisted of listening to trials with no figures, easy-to-detect figures, difficult-to-detect figures and one practice block of fifty mixed trials. For the main experiment, the value of coherence and duration was displayed before the start of each block and participants were instructed to press a button as soon as they heard a figure pop out of the random background (for the brief figures used here, these sounded like a 'warble' in the on-going random pattern). Feedback was provided. Blocks with different values of coherence and duration were presented in a pseudorandom order. The participants self-paced the experiment and the study lasted approximately an hour and a half. The procedure was identical across all experiments.

## Analysis

Participants' responses were measured in terms of sensitivity (d prime, or d'). We also report hit rates for certain conditions as mean ± one standard error.

## Apparatus

All stimuli were created online using MATLAB 7.5 software (The Mathworks Inc., Natick, MA) at a sampling rate of 44.1 kHz and 16 bit resolution. Sounds were delivered diotically through Sennheiser HD555 headphones (Sennheiser, Germany) and presented at a comfortable listening level of 60–70 dB SPL (self adjusted by each listener). Presentation of the stimuli was controlled using Cogent (http://www.vislab.ucl.ac.uk/cogent.php). Listeners were tested individually in an acoustically shielded sound booth. The apparatus was identical for all experiments.

## Temporal coherence model

The temporal coherence model consists of two distinct stages. The first stage takes the spectrogram of the stimulus as its input and simulates spectral analysis performed at the level of the cochlea and temporal integration at the level of the auditory cortex (*Chi et al., 2005*; *Elhilali and Shamma, 2008*; *Elhilali et al., 2009a*; *Shamma et al., 2011*). This temporal integration is achieved through a bank of bandpass filters that are tuned to different physiologically plausible parameters that capture the rich variety of spectrotemporal receptive fields (STRFs) found in A1. This is realized by using a range of physiologically valid temporal modulation rates (from slow to fast: 2–32 Hz) and spectral resolution scales (narrow to broad: 0.125 to 8 cycles per octave).

The next level of the model incorporates a 'coherence analysis' stage which computes a 'windowed' correlation between each pair of channels by taking the product of the filter outputs corresponding to the different channels. A dynamic coherence matrix which consists of the cross-correlation values as a function of time is thus obtained. The off-diagonal elements of the matrix indicate the presence of coherence across different channels, that is, positively correlated activity and offer insight into the perceptual representation of the stimulus.

## Procedure

The temporal coherence model was run for a range of temporal modulation rates: 2.5, 5, 10 and 20 Hz for experiments 1, 4, 5, and 6a and 5, 10, 20 and 40 Hz for experiment 3 respectively. Additionally, we used a rate of 3.33 Hz corresponding to the rate of presentation of 300 ms white noise segments in

experiment 6b. These rates cover the range of physiological temporal modulation rates observed in the auditory cortex. A single spectral resolution scale of 8 cycles per octave (corresponding to the bandwidth of streaming; 4 cycles per octave for experiment 4b where larger frequency steps are required to extract a ramped figure) was used.

The analysis was conducted by entering the SFG stimulus for each experimental condition to the input stage of the model. For experiments 1 and 3, the entire stimulus duration was fed to the model input and for the remaining experiments a stimulus without the pre- and post-figure chords was entered. This was based on the prediction that the background chords before and after the figure onset contribute little to the cross-correlation matrix unlike the chords comprising the figure. The simulations were performed separately for the stimuli containing a figure and without a figure and repeated across 500 iterations. To establish differences between the resultant coherence matrices, we computed the maximum value of the cross-correlation across all time points. This spectral decomposition helps us to examine which channels are correlated with each other (hence, the channels with repeating figure components could possibly be bound together as one object, or the 'figure'), and not significantly correlated with each other (hence, the channels with random correlation between channels may not be perceived as a single object). The difference in the average values of the maxima between the figure and the ground stimuli was calculated as the model response and plotted like the psychophysical curves to obtain model responses (see *Figure 3* and *Figure 3—figure supplement 1*).

## Acknowledgements
We thank Deborah Williams, Aiysha Siddiq, Nicolas Barascud, Madhurima Dey and Michael Savage for data collection.

## Additional information

### Funding

| Funder | Grant reference number | Author |
| --- | --- | --- |
| Wellcome Trust | WT091681MA | Sundeep Teki, Timothy D Griffiths |
| Wellcome Trust | 093292/Z/10/Z | Maria Chait |
| National Institutes of Health | R01 DC 07657 | Shihab Shamma |
| Deafness Research UK | | Maria Chait |

The funders had no role in study design, data collection and interpretation, or the decision to submit the work for publication.

### Author contributions
ST, MC, SK, Conception and design, Acquisition of data, Analysis and interpretation of data, Drafting or revising the article; SS, TDG, Conception and design, Analysis and interpretation of data, Drafting or revising the article

### Ethics
Human subjects: Experimental procedures were approved by the research ethics committee of University College London (Project ID number: 1490/002), and written informed consent was obtained from each participant.

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
