## [Decision Letter]

Thank you for sending your work entitled “Segregation of complex acoustic scenes based on temporal coherence” for consideration at *eLife*. Your article has been favorably evaluated by a Senior editor and 3 reviewers, one of whom is a member of our Board of Reviewing Editors, and one of whom, Mitchell Steinschneider, wants to reveal his identity.

The Reviewing editor and the other reviewers discussed their comments before we reached this decision, and the Reviewing editor has assembled the following comments to help you prepare a revised submission.

This is a well-written psychoacoustical study demonstrating that subjects are able to “hear out” a sequence of tones embedded in a complex acoustic environment of randomly presented tones. In contrast to other studies of this kind, there are no protective spectral regions where the background tones are omitted. The authors demonstrate that after initial training, most subjects are able to segregate the foreground sound pattern (one sound source) from the background sound source. Studies were appropriately performed with adequate controls. Indeed, it is remarkable that segregation of the foreground tone sequence still occurs despite introduction of interleaved bursts of white noise between the foreground tones. The duration of the white noise could extend up to 500 ms. These findings are clearly beyond the segregation capacities of core auditory cortex and the authors rightfully suggest non-classical auditory cortical processing stations as being the most likely generators of this percept. Further, the authors concisely provide modeling data and evidence that the temporal coherence model is the most appropriate model for explaining how this foreground/background segregation develops.

While the psychoacoustic results are interesting, the reviewers were disappointed by the integration between data and modeling.

1) First, the motivation for each modification of stimulus parameters should be explained more clearly. What would their model predict for the manipulation?

2) Second, it is unclear why the model simulations are not shown for all experimental manipulations, but only for number of tones and frequency components. It would be nice to see what the model predicts for the ramped SFG.

---

## [Author Response]

*1) First, the motivation for each modification of stimulus parameters should be explained more clearly. What would their model predict for the manipulation*?

We agree with this comment and we have restructured the manuscript to better integrate the psychophysics and the modeling, and to provide a clear motivation for the simulation parameters for each experiment. See the Results section for a better description of the modeling, starting “The temporal coherence model is based on the idea that a perceptual “stream” emerges when a group of (frequency) channels are coherently activated against the backdrop of other uncorrelated channels (Shamma et al, 2010).”

*2) Second, it is unclear why the model simulations are not shown for all experimental manipulations, but only for number of tones and frequency components. It would be nice to see what the model predicts for the ramped SFG*.

In addition to the modeling of the first (“basic SFG”) experiment (Figure 3), we have now performed and included simulations for the remaining experiments (new Figure 3—figure supplement 1) demonstrating results qualitatively similar to those in Figure 3.